# Characteristics and actions in high-risk COPD in unstable patients: The EPOCONSUL audit

Myriam Calle Rubio[1], Bernardino Alcázar-Navarrete[2], [3], José Luis López-Campos[4, 5],
Marc Miravitlles[6], Juan José Soler-Cataluña[7], [8], Manuel E. Fuentes Ferrer[9],
Juan Luis Rodríguez Hermosa[10], *

1 Department of Pulmonology, Hospital Clínico San Carlos, Department of Medicine, School of Medicine, Universidad Complutense de Madrid, Instituto de Investigación Sanitaria del Hospital Clínico San Carlos (IdISSC). CIBER de Enfermedades Respiratorias (CIBERES), Madrid, Spain, 2 Department of Pulmonology, Hospital Virgen de las Nieves, Granada, Spain, 3 IBS-Granada, Department of Medicine, Universidad de Granada, Granada, Spain, 4 Respiratory Disease Medical-Surgical Unit, Instituto de Biomedicina de Sevilla (IBiS), Hospital Universitario Virgen del Rocío/Universidad de Sevilla, Seville, Spain, 5 CIBER de Enfermedades Respiratorias (CIBERES), Instituto de Salud Carlos III, Madrid, Spain, 6 Department of Pulmonology, Hospital Universitari Vall d'Hebron, Vall d'Hebron Institut de Recerca (VHIR), CIBER de Enfermedades Respiratorias (CIBERES), Barcelona, Spain, 7 Department of Pulmonology, Hospital Arnau de Vilanova-Lliria, Valencia, Spain, 8 Department of Medicine, Universitat de València. CIBER de Enfermedades Respiratorias (CIBERES). Instituto de Salud Carlos III, Madrid, Spain, 9 Research Unit, Department of Preventive Medicine, Hospital Universitario Nuestra Señora de Candelaria, Santa Cruz de Tenerife, Spain, 10 Department of Pulmonology, Hospital Clínico San Carlos, Department of Medicine, School of Medicine, Universidad Complutense de Madrid, Instituto de Investigación Sanitaria del Hospital Clínico San Carlos (IdISSC), Madrid, Spain

* jlrhermosa@yahoo.es

## Abstract

### Objective:

To assess the clinical characteristics of high-risk COPD patients considered not stable for having had moderate or severe exacerbations of COPD in the three months prior to the audited review visit based on information extracted from the medical record documenting health interactions prior to the visit, and to analyse the therapeutic measures adopted at the follow-up visit.

### Methods

This analysis used data from the EPOCONSUL audit, which evaluated outpatient care provided to COPD patients in respiratory clinics in Spain. This analysis included patients with a high-risk level of COPD and assessed patient non-stability at the audited visit defined based on moderate or severe exacerbations in the last three months that were reported at the follow-up visit. Results: 2008 high-risk patients were analysed. 30.1% of patients were considered unstable at visit. Factors associated with non-stability are dyspnoea (MRC-m) ≥2 (OR 1.5, 95% CI 1.18–1.92; p = 0.001), chronic bronchitis criteria (OR 1.61, 95% CI 1.15–2.25; p = 0.005), use of inhaled triple therapy (OR 1. 31, 95% CI 1.06–1.61; p = 0.010), use of oral therapies for COPD

**Data availability statement:** All relevant data are within the manuscript and its Supporting Information files.

**Funding:** This study has been promoted and sponsored by the Spanish Society of Pneumology and Thoracic Surgery (SEPAR). This study has been promoted and sponsored by the Spanish Society of Pneumology and Thoracic Surgery (SEPAR). The funders had no role in the study design, data collection and analysis, decision to publish or preparation of the manuscript.

**Competing interests:** JLRH has received speaker fees from Bial, Boehringer Ingelheim, CSL Behring, GlaxoSmithKline, Zambon and Grifols, and consulting fees from Bial. MM has received speaker fees from AstraZeneca, Boehringer Ingelheim, Chiesi, Cipla, GlaxoSmithKline, Menarini, Kamada, Takeda, Zambon, CSL Behring, Specialty Therapeutics, Janssen, Grifols and Novartis, consulting fees from AstraZeneca, Atriva Therapeutics, Boehringer Ingelheim, Chiesi, GlaxoSmithKline, CSL Behring, Inhibrx, Ferrer, Menarini, Mereo Biopharma, Spin Therapeutics, Specialty Therapeutics, ONO Pharma, Palobiofarma SL, Takeda, Novartis, Novo Nordisk, Sanofi, Zambon and Grifols and research grants from Grifols. JLLC has received honoraria for lecturing, scientific advice, participation in clinical studies or writing for publications for: AstraZeneca, Bi-al, Boehringer Ingelheim, Chiesi, CSL Behring, Ferrer, Gebro, GlaxoSmithKline, Grifols, Menarini, Megalabs, Novartis and Rovi. JJSC has received speaker fees from AstraZeneca, Bial, Boehringer Ingelheim, Chiesi, FAES, GlaxoSmithKline, Menarini and Novartis, and consulting fees from Bial, Chiesi and GSK, and grants from GSK. BAN reports grants and personal fees from GSK, personal fees and non-financial support from Boehringer Ingelheim, personal fees and non-financial support from Chiesi, non-financial support from Laboratorios Menarini, grants, personal fees and non-financial support from AstraZeneca, personal fees from Gilead, personal fees and non-financial support from MSD, personal fees from Laboratorios BIAL, personal fees from Zambon, outside the submitted work; in addition, BAN has a patent P201730724 issued. MCR has received speaker fees from AstraZeneca,

(OR 1.68, 95% CI 1.23–2.28, p = 0.001), use of long-term oxygen therapy (OR 1.36, 95% CI 1.07–1.73, p = 0.010), no follow-up in a specialist COPD clinic (OR 1.44, 95% CI 1.11–1.87, p = 0.006). In 10.1% of the patients considered not stable, because at the medical visit they were referred to have had moderate or severe exacerbations in the last three months, no action was taken at the visit and in 56% there was no change in COPD pharmacological treatment. Triple therapy was the most commonly prescribed therapy (68% in non-stable patients). Twenty-five percent of patients on triple inhaled therapy are also prescribed oral therapy.

## Conclusions

One third of patients with high-risk COPD report exacerbations requiring treatment with antibiotics and/or systemic corticosteroids in the previous three months at the medical visit; and in more than half of these patients no changes in pharmacological treatment are made at the visit.

## Introduction

Chronic obstructive pulmonary disease (COPD) is a disease characterized by frequent exacerbations, which are responsible for increased morbidity and mortality [1]. Clinical practice guidelines for COPD identify minimizing the risk of future exacerbations as a key therapeutic objective [2,3] considering therapeutic success as achieving disease control. Clinical control of COPD is a measure based on two components: clinical impact (degree of dyspnea, use of rescue therapy, daily physical activity, sputum color) and stability (exacerbations) over time [4]. Previous studies have shown that poor clinical control for having exacerbations is a predictor of future exacerbations [5–7], with an increased risk of having a moderate or severe exacerbation in the following 6 months between 3 and 4 times [8].

In COPD, clinical practice guidelines identify risk stratification of COPD patients as a key step in planning therapeutic interventions and pharmacological decision making [2,3]. Lack of stability, due to persistent exacerbations on follow-up, is one of the main reasons for more careful patient assessment and more proactive therapeutic intervention by the clinician. In higher risk patients with exacerbations, a more individualized assessment is required, including the identification of features to guide therapeutic interventions. The identification of certain clinical conditions versus "treatable features" will aim to achieve better outcomes as responses to treatments are different. Thus, current evidence suggests a greater benefit from the addition of inhaled corticosteroids to bronchodilator therapy, depending on blood eosinophil concentration, frequency and severity of exacerbations and active smoking [9,10]. Other non-inhaled therapies in patients with poor control who persist with exacerbations despite optimal inhaled therapy may also be considered, the indications for which are determined by the presence of specific characteristics or conditions based on the results of clinical trials that have evaluated their benefits [11–13]. However, despite these recommendations there is little knowledge about the interventions performed

Bial, Chiesi, CSL Behring, GlaxoSmithKline, Menarini, and Grifols, and consulting fees from GlaxoSmithKline and Bial. The funders had no role in the design of the study; in the collection, analyses, or interpretation of data; in the writing of the manuscript; or in the decision to publish the results.

in high-risk patients with persistent exacerbations and the use in clinical practice of these oral therapies for COPD and the factors associated with their prescription in COPD follow-up [14,15].

Our analysis aims to evaluate the characteristics of high-risk COPD patients who remain unstable due to COPD exacerbations and the therapeutic measures adopted.

## Methodology

The methodology of the EPOCONSUL audit has been reported previously [16,17]. In brief, the EPOCONSUL audit, sponsored by the Spanish Society of Pneumology and Thoracic Surgery (SEPAR), was designed to evaluate the outpatient care of patients with COPD in respiratory clinics in Spain as an observational, non-interventional, cross-sectional study.

SEPAR sent an official invitation to participate in the study to all Spanish respiratory units with outpatient respiratory clinics, according to the Ministry of Health registry and the SEPAR member registry. Investigators participating in 2021 EPOCONSUL are listed in S1 Appendix. The study inclusion was performed between 15 April 2021 and 31 January 2022. Recruitment was intermittent; each month, each investigator recruited the clinical records of the first 10 patients diagnosed with COPD seen at the respiratory outpatient clinic. The identified patients were then re-assessed to determine whether they met the inclusion/exclusion criteria described in S2 Appendix. The information collected in the audit was of a historical nature in terms of clinical data from the last consultation and previous consultations. The high-risk level was defined according to the Spanish COPD guideline criteria (GesEPOC) [2]: dyspnea grade≥2 (MRC-m) and/or 2 or more moderate exacerbations and/or ≥1 hospitalization and/or FEV1<50% predicted described in S3 Appendix. "Non-stability (non-stable patient) was defined on the basis of having moderate or severe exacerbations of COPD in the three months prior to the audited review visit. All data were verified by clinical investigators with access to the original documentation. Data were systematically extracted from medical records documenting health interactions prior to the audited visit to ensure completeness and consistency with records made at the audited visit." 4225 patients with COPD who were followed up in the respiratory clinics of 45 hospitals in Spain were studied. Of these, 2008 (47.5%) patients who met all the Spanish COPD guideline criteria recorded at the visit to define non-stability in the high-risk level were analyzed. The sampling procedure is shown in S1 Fig.

This analysis evaluated the characteristics of high-risk COPD patients who considered not stable, because they reported at the follow-up medical visit having exacerbations requiring antibiotic and/or corticosteroid treatment in the three months prior to the audited visit and the therapeutic measures taken in the audited visit. For this purpose, we analyzed socio-demographic variables (age, sex), clinical characteristics (smoking history, body mass index (BMI), comorbidity, cardiovascular disease, asthma, degree of dyspnea MRC, peripheral eosinophilia, exacerbating phenotype, criteria chronic bronchitis), resources (consultation model, level of complexity of the center), and actions taken (COPD treatments and diagnostic tests), as well as treatment changes made at the visit.

The protocol was approved by the Ethics Committee of the Hospital Clínico San Carlos (25/11/2020, Madrid, Spain; internal code 20/722-E). In addition, the ethics committee of each participating hospital evaluated and approved the study protocol, in accordance with current research legislation in Spain. Informed consent was not required due to the non-interventional nature of the study, the anonymization of data and the blinded assessment of clinical performance, as well as the fact that this was a clinical audit. The medical staff responsible for the respiratory outpatient clinic were not informed of the audit to avoid changes in usual clinical practice and to maintain blinding of the clinical performance assessment.

### Statistical analysis

Qualitative variables were summarized as frequency distribution and continuous variables as mean values and standard deviations. Continuous, non-normally distributed variables were expressed as medians and interquartile ranges (IQR). The association between qualitative variables was performed using the chi-square test, the comparison of means between quantitative variables and binary outcome variables was assessed using the Student's T-test, and the non-parametric Mann-Whitney U test was used in the case of continuous non-normally distributed variables.

A multivariable logistic regression model, using cluster-robust standard errors to take into account patients tested within the same hospital, was fitted in order to identify factors associated with patient non-stability, defined as having had moderate or severe exacerbations during the three months prior to the audited visit. Adjusted odds ratios (OR) and their 95% confidence intervals are shown. Independent variables with a p-value of $p < 0.05$ in the univariate analysis and/or those considered clinically relevant were added to the model. Statistical significance was assumed as $p < 0.05$. In the analysis of oral therapies in high-risk patients, a series of multivariate logistic regression models using cluster-robust standard errors were fitted to estimate the adjusted effect of oral treatments over clinical outcome variables (Chronic bronchitis, frequent Exacerbator, history of COPD hospitalization, chronic bronchial infection, peripheral eosinophilia≤100 cells/μL and FEV1 less than 50%). All analyses were performed using Stata software version 16 (StataCorp LLC, College Station, TX, USA).

## Results

### Study population

Of the total 2008 patients analyzed, 1,471 (73.3%) were men and the mean age was 70.2 (9.0) years. The mean FEV1% predicted (%) was 46.9 (16.6). Characteristics are shown in Table 1.

### Non-stability and factors associated

Of the total population of patients classified as high risk, 30.1% were considered unstable for having documented moderate or severe exacerbations in the medical record for the previous three months, and 69.9% were considered stable. Clinical characteristics according to not stability in visit are shown in Table 1. Factors associated with non-stability of patient in visit are shown in Table 2 (multivariate logistic regression model): dyspnea (MRC-m) ≥2 (OR 1.5, 95% CI 1.18–1.92; p=0.001), chronic bronchitis criteria (OR 1.61, 95% CI 1.15–2.25; p=0.005), use of inhaled triple therapy (OR 1.31, 95% CI 1.06–1.61; p=0. 0.010), use of oral therapies for COPD (OR 1.68, 95% CI 1.23–2.28, p=0.001), use of long-term oxygen therapy (OR 1.36, 95% CI 1.07–1.73, p=0.010), no follow-up in a specialist COPD clinic (OR 1.44, 95% CI 1.11–1.87, p=0.006) and not being treated in a university center (OR 0.55, 95% CI 0.33–0.90, p=0.018). Fig 1 shows the distribution of some characteristics to be taken into account in high-risk COPD patients with frequent exacerbations who are considered unstable for having documented moderate or severe exacerbations in the medical record during the three months prior to the audited visit, such as peripheral eosinophilia≥300 cells/μL or the presence of asthma as comorbidity as features to be considered for specific therapies targeting Th2 inflammation.

**Table 1. Characteristics of high-risk patients according to stability at the visit.**

| High-risk level | All | Patients with stability | Patients without stability | p |
|---|---|---|---|---|
| N = 2008 | N = 2008 | n = 1404 (69.9%) | n = 604 (30.1%) | |
| Demographic and clinical characteristics | | | | |
| Gender (male), n (%) | 1471 (73.3) | 1027 (73.1) | 444 (73.5) | 0.886 |
| Age (years), m (SD) | 70.2 (9.0) | 69.8 (8.8) | 71.2 (9.5) | 0.002 |
| ≤ 55, n (%) | 108 (5.4) | 77 (5.5) | 31 (5.1) | 0.035 |
| 56–69, n (%) | 823 (41) | 600 (42.8) | 223 (36.9) | |
| ≥ 70, n (%) | 1075 (53.6) | 725 (51.7) | 350 (57.9) | |
| Smoking status | | | | 0.489 |
| Current smokers, n (%) | 472 (23.5) | 324 (23.1) | 148 (24.5) | 0.078 |
| Ex-smokers, n (%) | 1536 (76.5) | 1080 (76.9) | 456 (75.5) | |
| Pack-years, m (SD) | 51.4 (23.6) | 50.8 (23.8) | 52.8 (23.1) | |
| BMI kg/m2, m (SD) | 27.6 (5.8) | 27.7 (6) | 27.3 (5.3) | 0.489 |
| ≤ 21, n (%) | 472 (23.5) | 324 (23.1) | 148 (24.5) | 0.078 |
| ≥ 30, n (%) | 1536 (76.5) | 1080 (76.9) | 456 (75.5) | |
| Comorbidity | | | | 0.007 |
| Charlson index, median, IQR | 2 (1-3) | 2 (1-3) | 2 (1-3) | 0.054 |
| Charlson index ≥3, n (%) | 594 (29.6) | 397 (28.3) | 197 (32.6) | |
| Cardiovascular disease, n (%) | 818 (40.7) | 533 (38) | 285 (47.2) | <0.001 |
| Asthma, n (%) | 324 (21.1) | 234 (22) | 90 (18.9) | 0.170 |
| Rinithis/nasal polyposis, n (%) | 79 (3.9) | 54 (3.9) | 25 (4.1) | 0.761 |
| Dyspnea (MRC-m) ≥2, n (%) | 1529 (76.1) | 1020 (72.6) | 509 (84.3) | <0.001 |
| CAT mayor de 10, n (%) | 649 (32.3) | 407 (29) | 242 (40.1) | <0.001 |
| Chronic bronchitis criteria, n (%) | 782 (38.9) | 474 (33.8) | 308 (51) | <0.001 |
| Peripheral eosinophilia, median (IQR) | | | | 0.721 |
| < 100 mm3, n (%) | 371 (27.5) | 256 (27.9) | 115 (26.6) | |
| < 150 mm3, n (%) | 536 (39.7) | 358 (39) | 178 (41.2) | |
| ≥ 300 mm3, n (%) | 311 (23.1) | 212 (23.1) | 99 (22.9) | |
| Post-FEV$_1$, % predicted, m, (SD) | 46.9 (16.6) | 47.5 (17.0) | 45.6 (15.6) | 0.017 |
| < 50%, n (%) | 1242 (61.8) | 857 (61) | 385 (63.7) | 0.117 |
| 50–79%, n (%) | 688 (34.3) | 207 (14.7) | 207 (34.3) | |
| ≥ 80%, n (%) | 78 (3.9) | 66 (4.7) | 12 (2) | |
| % ≥ 2 moderate exacerbations in the last year, n (%) | 483 (24.1) | 217 (15.5) | 266 (44) | <0.001 |
| % ≥ 1 hospital admissions in the last year, n (%) | 551 (27.4) | 226 (16.1) | 325 (53.8) | <0.001 |
| BODE value, m, (SD) | 4.3 (1.9) | 4.1 (1.9) | 4.6 (1.9) | 0.005 |
| Inhaled Therapies, n (%) | | | | <0.001 |
| LAMA or LABA | 65 (3.3) | 57 (4.1) | 8 (1.3) | |
| LAMA-LABA combination | 605 (30.6) | 470 (34.1) | 135 (22.6) | |
| LABA+ ICS combination | 120 (6.1) | 77 (5.6) | 43 (7.2) | |
| Triple therapy (LAMA+LABA+CSI) | 1184 (60) | 773 (56.1) | 411 (68.8) | |
| Roflumilast, n (%) | 72 (3.6) | 32 (2.3) | 40 (6.6) | <0.001 |
| Mucolitics, n (%) | 112 (5.6) | 54 (3.8) | 58 (9.6) | <0.001 |
| Methylxanthines, n (%) | 44 (2.2) | 26 (1.9) | 18 (3) | 0.113 |
| Macrolides, n (%) | 139 (6.9) | 73 (5.2) | 66 (10.9) | <0.001 |
| Long-term oxygen therapy, n (%) | 635 (31.6) | 380 (27.1) | 255 (42.2) | <0.001 |
| Home ventilation, n (%) | 185 (9.2) | 114 (8.1) | 71 (11.8) | 0.010 |
| Respiratory rehabilitation, n (%) | 336 (16.7) | 223 (15.9) | 113 (18.7) | 0.120 |

*(Continued)*

**Table 1.** (Continued)

| High-risk level<br>N = 2008 | All<br>N = 2008 | Patients with stability<br>n = 1404 (69.9%) | Patients without stability<br>n = 604 (30.1%) | p |
|---|---|---|---|---|
| Resources in care | | | | |
| Attended in specialized COPD outpatient clinic, n (%) | 990 (49.5) | 724 (51.8) | 266 (44.1) | 0.002 |
| Respiratory care follow-up (years) median, IQR | 7.1 (4.7) | 7.0 (4.6) | 7.3 (4.9) | 0.189 |
| Scheduled follow-up visits, n (%) | | | | <0.001 |
| <6 months | 947 (50.1) | 563 (43) | 384 (66.1) | |
| ≥6 months | 943 (49.9) | 746 (57) | 197 (33.9) | |
| Level of complexity of hospital, n (%) | | | | 0.127 |
| Secondary | 359 (17.9) | 239 (17) | 120 (19.9) | |
| Tertiary | 1649 (82.1) | 1165 (83) | 484 (82.1) | |
| University Hospital, n (%) | 1693 (84.3) | 1228 (87.5) | 465 (77) | <0.001 |
| Availability of nursing consultation, n (%) | 1216 (60.6) | 884 (63) | 332 (55) | <0.001 |

Data are presented as mean (SD) or median (P25-75). Dichotomous variables are expressed as *n* and percentage. Non-stable patient: having COPD exacerbations since last visit; BMI, body mass index; mMRC, modified Medical Research Council; CAT, COPD Assessment Test; FEV1%: post-bronchodilator FEV1 percent predicted; BODE: body mass index, airflow obstruction, dyspnea, and exercise capacity; GesEPOC, Spanish National Guideline for COPD; LABA, long-acting beta-2 agonists; LAMA, long-acting antimuscarinic agents; CSI, Inhaled corticosteroids.

**Table 2. Factors related to Non-stability of COPD. "Multivariable logistic model".**

| Clinical characteristics | OR | [CI 95% (OR)] | p |
|---|---|---|---|
| Age (years) | 1.01 | 0.99–1.02 | 0.088 |
| Charlson index <3 (ref) | 1 | 0.78–1.21 | 0.831 |
| Charlson index ≥3 | 0.97 | | |
| Dyspnea (MRC-m) <2 (ref) | 1 | 1.18–1.92 | 0.001 |
| Dyspnea (MRC-m) ≥2 | 1.51 | | |
| Post-FEV$_1$, % predicted | 0.99 | 0.99-1.00 | 0.831 |
| Chronic bronchitis criteria, No (ref) | 1 | 1.15–2.25 | 0.005 |
| Yes | 1.61 | | |
| Triple therapy (LAMA+LABA+ICS), No (ref) | 1 | 1.06 −1.61 | 0.010 |
| Yes | 1.31 | | |
| Oral therapies for COPD, No (ref) | 1 | 1.23 −2.28 | 0.001 |
| Yes | 1.68 | | |
| Long-term oxygen therapy, No (ref) | 1 | 1.07–1.73 | 0.010 |
| Yes | 1.36 | | |
| Care process | | | |
| Treated in specialized COPD outpatient clinic, Yes (ref) | 1 | 1.11-1.87 | 0.006 |
| Not | 1.44 | | |
| University Hospital, No (ref) | 1 | 0.33–0.90 | 0.018 |
| Yes | 0.55 | | |
| Scheduled follow-up visits, <6 months (ref) | 1 | 0.36- 0.64 | <0.001 |
| ≥6 months | 0.48 | | |

Not stability was defined by having exacerbations of COPD since the last visit.

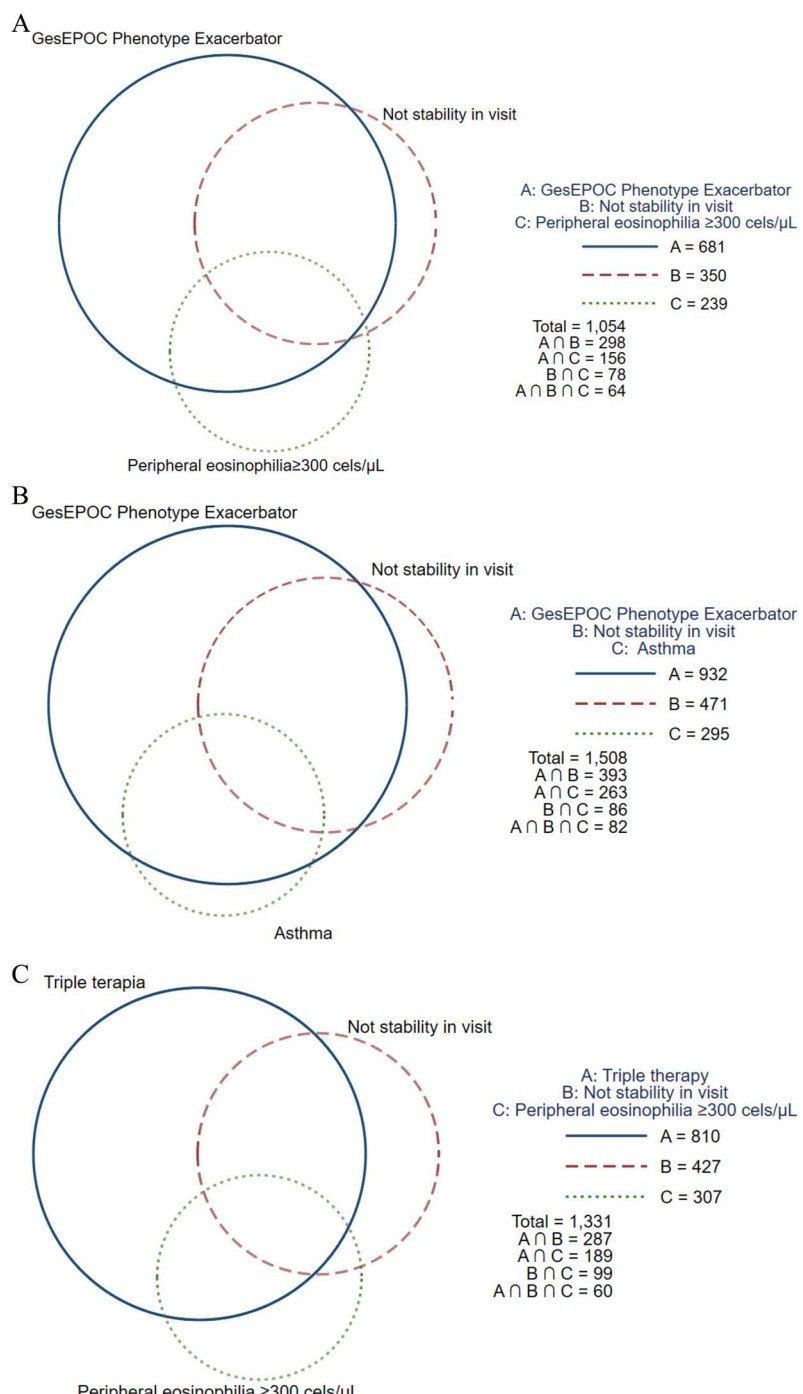

**Fig 1. Distribution of characteristics in high risk COPD.** A: Distribution of caracteristics (phenotype exacerbator, not stability in visit, peripheral eosinophilia≥300 cels/μL) in high risk COPD. B: Distribution of caracteristics (phenotype exacerbator, not stability in visit, asthma) in high risk COPD. C: Distribution of caracteristics (not stability in visit, peripheral eosinophilia≥300 cels/μL, inhaled triple therapy) in high risk COPD. The size of each circle is proportional to the number of individuals in each group. Abbreviation: GesEPOC: Spanish National Guideline for COPD.

## Pharmacological therapies in high-risk COPD patients

Of the 1974 high-risk patients studied for whom COPD treatment was recorded at the visit, 85.5% were treated with inhaled therapies only and 14.5% were treated with both oral and inhaled COPD therapies. Fig. 2 shows the distribution of COPD pharmacological treatments according to the stability achieved of patient in audited visit (the existence or not of exacerbations during the three months prior to the audited visit. Oral COPD therapies were more frequently prescribed in patients with persistent exacerbations at the visit (21.9% in non-stability versus 11.3% in stability, p<0.001).

Fig 3 show the distribution of inhaled and oral therapies in high-risk patients. Triple therapy was the leading treatment in high-risk patients (60%), followed by dual bronchodilator therapy (30.6%). Among oral therapies, maintenance macrolides (48.4%), mucolytics (38.7%) and roflumilast (25.1%) were the most commonly used, while methylxanthines were the least used (15.3%).

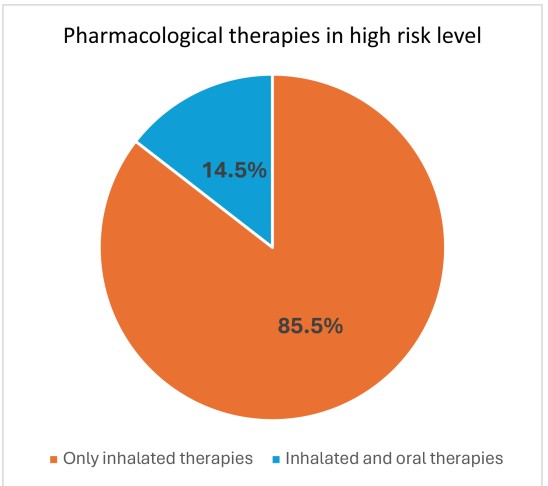
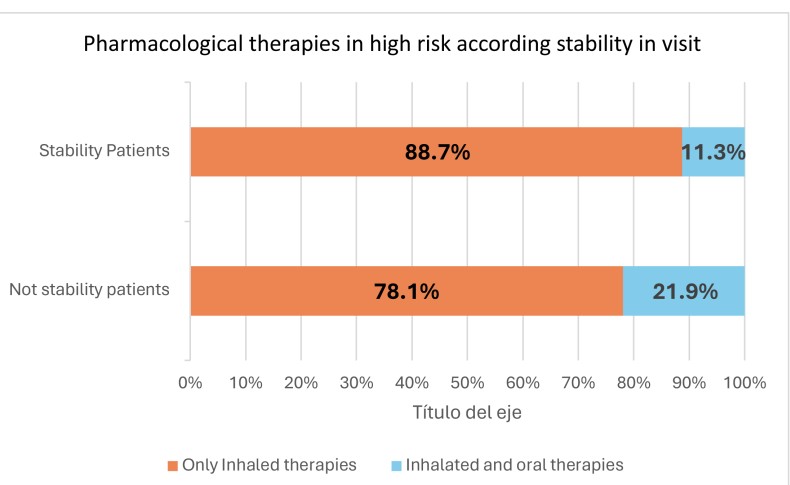

**Fig 2. Distribution of COPD drug therapies in high-risk COPD (a) according stability in visit(b).** Differences stability versus non-stability at visit P<0.001.

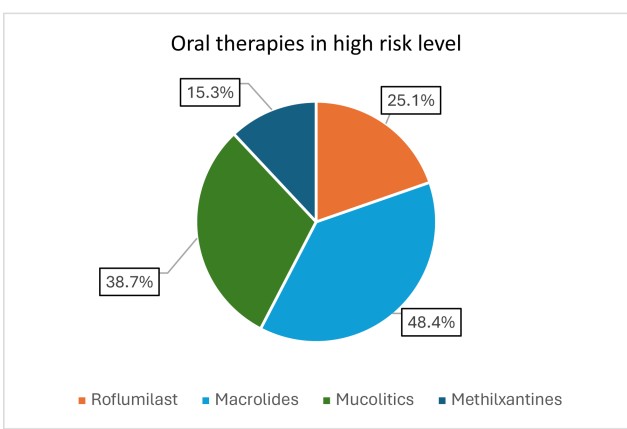
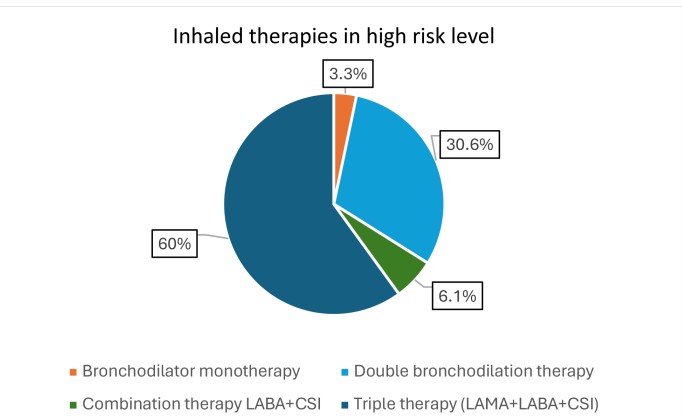

**Fig 3. The distribution of oral (a) and inhaled (b) therapies in high risk COPD.** Abbreviations: LABA: long-acting beta-2 agonists; LAMA: long-acting antimuscarinic agents; CSI: Inhaled corticosteroids.

The distribution of inhaled and oral therapies in high-risk patients is summarized in Fig. 4. 25% of patients on triple inhaler therapy were also prescribed oral therapies, the most common being chronic macrolides (9.9%) and mucolytics (7.1%). 8.2% of patients on dual bronchodilator therapy also received oral therapy, the most common being mucolytics (2.9%).

## Oral therapies and associated factors

Oral therapies for COPD were prescribed in 14.5% of high-risk patients. Table 3 shows, by multivariate logistic regression model, the adjusted effect of oral treatments with a series of clinical factors. The use of roflumilast, adjusted for other oral treatments, was associated with Chronic bronchitis (OR 3.6, 95% CI 1.7–7.4; p = 0.001), frequent exacerbator (OR 4.0, 95% CI 1.1–14.5; p = 0.035), history of COPD hospitalisation (OR 2.5, 95% CI 1.3–4.8; p = 0.006) and chronic bronchial infection (OR 2.1, 95% CI 1.1–3.9; p = 0.016). Chronic macrolides use, adjusted for other therapy, is associated with chronic bronchitis (OR 2.6, 95% CI 1.4–4.8; p = 0.002), FEV1 less than 50% predicted (OR 1.7, 95% CI 1.0–3.1;

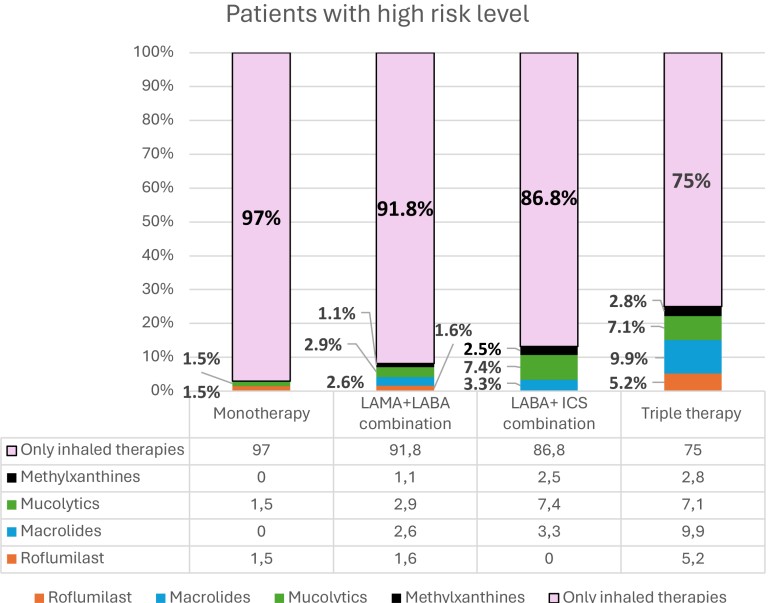

**Fig 4. Distribution of oral therapies for COPD in high-risk patients on inhaled therapy.** Abbreviations: LABA: long-acting beta-2 agonists; LAMA: long-acting antimuscarinic agents; CSI: Inhaled corticosteroids.

**Table 3. Factors related to oral therapies for COPD. "Multivariable logistic model".**

|  | Chronic bronchitis criteria | GesEPOC Phenotype Exacerbator | ≥1 hospital admissions for COPD in last year | Post-FEV1 < 50% predicted | Chronic bronchial infection | Peripheral eosinophilia ≤100 mm3 |
|---|---|---|---|---|---|---|
|  | OR (CI 95%) | OR (CI 95%) | OR (CI 95%) | OR (CI 95%) | OR (CI 95%) | OR (CI 95%) |
| **Roflumilast** | 3.6 (1.7-7.4)* | 4.0 (1.1-14.5)* | 2.5 (1.3-4.8)* | 1.3 (0.5-3.1) | 2.1 (1.1-3.9)* | 0.6 (0.2-2.0) |
| **Macrolides** | 2.6 (1.4-4.8)* | 1.8 (0.5-5.7) | 1.3 (0.8-2.2) | 1.7 (1.0-3.1)* | 4.5 (2.5-8.3)* | 0.3 (0.1-0.8)* |
| **Mucolitics** | 3.8 (1.7-8.4)* | 1.8 (0.8-3.9) | 1.8 (1.0-3.1)* | 0.7 (0.3-1.5)* | 2.5 (1.4-4.6)* | 1.1 (0.5-2.4) |
| **Methilxantines** | 1.7 (0.8-3.7) | 1.9 (0.6-5.8) | 0.8 (0.3-1.7) | 2.5 (1.1-5.7)* | 0.7 (0.2-2.0) | 0.6 (0.2-1.4) |

GesEPOC, Spanish National Guideline for COPD; Chronic bronchial infection: isolates of the same potentially pathogenic microorganism in respiratory samples. *p < 0.05.

p = 0.048) and peripheral eosinophilia ≤100 cells/μL (OR 0.3, 95% CI 0.1–0.8; p = 0.013). The use of mucolytics is associated with chronic bronchitis (OR 3.8, 95% CI 1.7–8.4; p = 0.001), history of COPD hospitalisation (OR 1.8, 95% CI 1.0–3.1; p = 0.026) and chronic bronchial infection (OR 2.5, 95% CI 1.4–4.6; p = 0.001). Use of methylxanthines, adjusted for other therapies, was associated with FEV1 less than 50% (OR 2.5, 95% CI 1.1–5.7; p = 0.021). The distribution of the different oral therapies according to characteristics associated with their use is shown in S1 Table.

**Actions taken at high risk and no-stability at visit**

Among all high-risk patients who at the visit were considered not stable for having documented moderate or severe exacerbations in the medical record during the three months prior to the audited visit, 10.1% of the patients no action was taken at the visit and in 56% of the visits of non-stable patients no change in COPD treatment was made. Table 4 shows the actions taken during the visit in patients with a high-risk level who were considered not stable at the visit. Requesting a test was the most common action taken at the follow-up visit (56%), with pulmonary function tests (in 41.9%) and imaging (41.9%) being the most common. A change in COPD treatment was made during the visit in 33.9% of patients, with an escalation (46.6%) or change to a similar regimen (46.1%). Factors associated with no change in therapy during the visit in high risk level not stability patients are shown in Table 5. Factors associated with no action taken at the visit in high-risk patients who were not stable due to persistent exacerbations documented in the medical record are shown in S2 Table.

## Discussion

This study provides novel information on characteristics of high-risk COPD patients who remain unstable at the follow-up visit due to COPD moderate or severe exacerbations during the three months prior to the audited visit and the therapeutic measures adopted. It also analyzes factors associated with non-stability, and drug prescription patterns of oral therapies in high-risk COPD, using real data from a clinical audit conducted in Spain.

The results of our analysis show that almost one-third of patients with high-risk COPD were considered unstable at the visit because they reported having suffered some exacerbation at the follow-up medical visit. Factors associated with non-stability were clinical characteristics such as the presence of chronic bronchitis or a higher degree of dyspnea, and the fact of not having been seen in a specialized COPD clinic or university center. In more than half of the audited visits of

**Table 4. Actions taken during the visit in high risk level not stability in visit.**

| n = 604 | |
|---|---|
| No action, n (%) | 61 (10.1) |
| Only Testing was requested, n (%) | 336 (56) |
| Only change in COPD treatment was made, n (%) | 15 (2.5) |
| Change of treatment and request for test was made, n (%) | 189 (31.4) |
| Change performed, n (%) | |
| Scaling (increased or added) | 95 (46.6) |
| De-escalate (decrease or remove) | 15 (7.4) |
| Changes to similar regimen | 94 (46.1) |
| Request for test in the visit, n (%) | |
| Pulmonary function test | 253 (41.9) |
| Imaging study | 253 (41.9) |
| Microbiological study | 152 (25.2) |
| Blood tests | 147 (24.3) |
| Cardiology study | 40 (6.6) |

Dichotomous variables are expressed as *n* and percentage.

**Table 5. Factors associated with no change in therapy during the visit in high risk level not stability patients.**

| n = 604 | Change in therapy in visit | No change in therapy in visit | p |
|---|---|---|---|
| | n = 203 (33.8%) | n == 397 (66.2%) | |
| **Clinical Characteristics** | | | |
| Gender (male), n (%) | 151 (74.4) | 290 (73) | 0.726 |
| Age (years), m (SD) | 70.1 (9.3) | 71.6 (9.6) | 0.080 |
| Current smokers, n (%) | 61 (30) | 86 (21.7) | 0.024 |
| Charlson index ≥3, n, (%) | 60 (29.6) | 136 (34.3) | 0.245 |
| Cardiovascular disease, n (%) | 95 (46.8) | 188 (47.4) | 0.897 |
| Asthma, n (%) | 33 (18.5) | 57 (19.3) | 0.847 |
| Dyspnea (MRC-m) ≥2, n (%) | 178 (87.7) | 327 (82.4) | 0.091 |
| CAT questionnaire > 10, n (%) | 98 (94.2) | 144 (78.3) | <0.001 |
| Chronic bronchitis criteria, n (%) | 111 (54.7) | 196 (49.4) | 0.218 |
| Chronic bronchial infection, n (%) | 58 (28.6) | 74 (18.6) | 0.005 |
| Post-FEV$_1$, % predicted, m (SD) | 45.5 (16.0) | 45.5 (15.3) | 0.982 |
| BODE value, m (SD) | 4.7 (1.6) | 4.5 (2.0) | 0.499 |
| ≥1 hospital admissions in the last year,n (%) | 109 (53.7) | 214 (53.9) | 0.961 |
| Peripheral eosinophilia, median (IQR) | 217.6 (151.2) | 240.5 (178.9) | 0.189 |
| ≤100 mm3, n (%) | 65 (45.1) | 112 (39.2) | 0.430 |
| 101-299 mm3, n (%) | 50 (34.7) | 104 (36.4) | |
| ≥ 300 mm3, n (%) | 29 (20.1) | 70 (24.5) | |
| GesEPOC Phenotype Exacerbator | 150 (84.3) | 243 (82.9) | 0.706 |
| LAMA or LABA | 2 (1) | 6 (1.5) | 0.890 |
| LAMA-LABA combination | 47 (23.4) | 87 (22.2) | 0.885 |
| LABA+ ICS combination | 13 (6.5) | 30 (7.7) | 0.957 |
| Triple therapy (LAMA+LABA+CSI), n (%) | 139 (69.2) | 269 (68.6) | 0.895 |
| Long-term oxygen therapy, n (%) | 80 (39.4) | 174 (43.8) | 0.300 |
| Home ventilation, n (%) | 26 (12.8) | 44 (11.1) | 0.533 |
| Public University Hospital, n (%) | 128 (63.1) | 333 (83.9) | <0.001 |
| Attended in specialized COPD outpatient clinic, n (%) | 82 (40.6) | 184 (46.3) | 0.180 |

Data presented as mean (SD) or number (percentage) or median (P25-75). mMRC, modified Medical Research Council; CAT, COPD assessment test; Post-FEV1%, post-bronchodilator FEV1 percent predicted; BODE, body mass index, airflow obstruction, dyspnea, and exercise capacity; GesEPOC, Spanish National Guideline for COPD; LABA, long-acting beta-2 agonists; LAMA, long-acting antimuscarinic agents; CSI, Inhaled corticosteroids.

COPD patients who were not stable, the therapeutic prescription was not changed. Triple therapy is the most prescribed pharmacological regimen in high-risk patients. A quarter of patients treated with triple inhaled therapy are also prescribed oral therapies for COPD, with their use being more frequent in patients who are not stable and who persist with exacerbations. Macrolides are the most commonly used (48.4%) among oral therapies for COPD, and the characteristics associated with their use were the presence of chronic bronchitis, chronic bronchial infection and severe airflow obstruction.

One of the major challenges in the management of COPD is the recurrence of exacerbations. These exacerbations not only increase morbidity and mortality [18,19] but also contribute to increased patient instability, reduced quality of life [20] and accelerated decline in lung function [21]. A proportion of patients appear to be more susceptible to exacerbations, with poorer quality of life and more aggressive disease progression. Observational studies assessing the degree of clinical control in COPD have shown significant variability in clinical control status, although a small proportion of patients remain persistently poorly controlled and are associated with an increased risk of death [5,6,7]. In our study, 30% of patients with high-risk COPD continued to have exacerbations at visit, with the presence of chronic bronchitis and a higher degree of

dyspnea in high-risk COPD identified as factors associated with failure to achieve stability. These findings are similar to those reported in previous studies, which showed that chronic bronchitis, high symptom burden, lower FEV1% predicted and a history of previous exacerbations were associated with an increased risk of exacerbations and mortality [22,23].

The correct assessment of COPD exacerbations will be crucial to assess risk, the degree of clinical control in COPD and to plan therapeutic interventions. Currently the identification of exacerbating patients is based on clinical records and/or patient recall [24]. However, these are not always reported. Thus, several studies suggest that about half of exacerbations are not reported, although they are associated with a worsening quality of life and an increased risk of subsequent hospitalisation [25,26]. On the other hand, it should be remembered that definitions of COPD exacerbations are mainly based on clinical presentation, which can be non-specific and lead to misdiagnosis. The use of biomarkers could provide a more objective and specific definition of COPD exacerbations and allow for a more accurate assessment of progression or response to treatment. Future research is needed to establish a better definition of COPD exacerbations through the use of biomarkers to improve diagnostic accuracy, more personalised treatment and potentially improve clinical outcomes and reduce healthcare costs.

Evidence from audit studies has suggested that outcomes in the management of COPD vary according to patient characteristics and the medical care received [27,28]. In our study, patients were 1.5 times more likely to have persistent exacerbations at follow-up if they did not attend a specialist COPD clinic. Specialized COPD consultation is a care model that has been developed in pulmonology units in recent years and was present in 62.2% of centers participating in the EPOCONSUL audit in 2021 [17]. Studies have shown that systematic care improves the quality of life and prognosis of patients with complex chronic diseases and reduces the cost of care [29]. These data support the need to adapt the COPD care model to ensure specialized units for patients with high complexity and frequent decompensation, where better care can be provided and resources optimized. However, it is important to recognize that there is a subgroup of COPD patients with frailty, associated social factors, lack of family support and multiple comorbidities [30, 31], which requires a different approach, focused on improving community health services and care planning, together with early identification of the most vulnerable patients [32].

In our study, in more than half of the patients with high-risk COPD who were not stable at the visit, no therapeutic intervention was performed. Therapeutic inertia is defined as recognition of the problem but lack of action [33]. A reality, the lack of modification of COPD treatment in follow-up visits, which was already noted in our country in the 2014 audit. In 77.5% of the audited visits of COPD patients in pulmonary follow-up visits, the therapeutic prescription was not changed [14]. In our analysis, few clinical characteristics were associated with the dependent variable no change in COPD treatment at the visit in a patient with high-risk COPD who reported exacerbations at the visit. In the multivariate logistic regression model, only being treated in a university hospital (OR 3.08, 95% CI 1.71–5.53; p < 0.001) was found as an associated factor, being less likely not to make changes in the active smoker (OR 0.63, 95% CI 0.44–0.90; p = 0.011), and the presence of chronic bronchial infection (OR 0.50, 95% CI 0.33–0.76; p = 0.001). Little is known about the reasons for the lack of action when treatment goals are not met. Studies have shown that physicians are willing to take actions when they deem them appropriate, but there is a disconnection between their perception and the actual degree of control of COPD in uncontrolled patients, suggesting that physicians do not feel the need to switch up therapies or increase doses if the patient is showing positive signs of improvement or maintenance [34]. Studies are needed to analyze the reasons for the non-implementation of clinical practice guidelines and the conditioning factors in clinical practice."

Analysis of medical records has shown that 23% to 28% of COPD patients do not receive guideline-recommended treatment [35], and only 33% to 60% of patients receive adequate treatment according to the GOLD recommendations [36]. However, it is important to bear in mind that a quarter of patients who persist with exacerbations are on quadruple therapy for COPD, and could therefore be considered to be on a drug treatment ceiling. In our analysis, being a cross-sectional observational study, prescription patterns may reflect disease severity rather than therapeutic effectiveness. Furthermore, it should be borne in mind that the clinical presentation of COPD is highly variable, and that other factors

associated with lack of clinical control and stability include associated comorbidities, adherence, among others. COPD is a complex and heterogeneous disease, which justifies the need for a more personalized approach in high-risk patients by identifying characteristics that guide us to offer more specific treatments to improve outcomes [1,2]. The lack of clinical stability in patients with frequent exacerbations is one of the main reasons for the need to adapt and personalize treatment in the follow-up of high-risk COPD patients. Early identification of exacerbation triggers and implementation of individualized treatment strategies are essential to improve disease control, reduce the frequency of acute episodes and optimize the long-term well-being of patients. In our study, high-risk patients with persistent exacerbations received a higher number of COPD therapies, as the vast majority of patients were treated with triple therapy and more than 20% were receiving oral COPD therapies, so they could be considered as patients in a therapeutic ceiling. However, it is noteworthy that, in this high-risk patient population who continue to experience exacerbations despite optimized treatment, other features were observed that could guide the choice of new therapeutic options targeting Th2 inflammation in COPD, such as peripheral blood eosinophil levels ≥300 cells/µL or the presence of asthma as an associated comorbidity [1,2].

In our population of unstable high-risk patients, 85% had a frequent exacerbator phenotype and 67% were on maintenance treatment with triple therapy. Of these, 23% had peripheral blood eosinophil counts ≥300 cells/µL, being a priori a population in which treatment optimization should be considered, assessing escalation to triple therapy and/or could benefit from biologic therapy. Evidence suggests that 20–40% of COPD patients have Th2 airway inflammation as measured by sputum eosinophilia [37,38]. This endotype is associated with increased risk of exacerbations. Elevated blood eosinophil counts (BEC), typically ≥300 cells/µL, serve as a practical biomarker, identify patients more likely to benefit from emerging biologics. Recent trials demonstrate that targeting T2 pathways produces significant reductions in exacerbations and improves lung function in COPD patients who persist with exacerbations despite optimization of inhaled therapy [38], reflecting a move toward precision medicine in this heterogeneous disease. In the near future, artificial intelligence-based models may help to identify this group of patients earlier in order to provide more personalized medicine for high-risk COPD patients. AI algorithms, including machine learning and deep learning, can analyze various types of data, such as medical records, patient-reported outcomes, environmental factors and genetic information, to predict exacerbations and disease progression. Models using support vector machines, boosting, and deep neural networks have shown promise in predicting COPD exacerbations with high accuracy [39].

The main strengths of our study are its national coverage and the comprehensive and systematic assessment of clinically relevant parameters together with resource and organizational aspects of clinical care. However, several methodological considerations must be taken into account in order to correctly interpret our results. The main limitation of the present study is its post hoc design. This audit focused on a cross-sectional assessment of the clinical history and a specific clinical visit, the last of which was carried out in the follow-up of his COPD in the pulmonology department. Another aspect to consider is the retrospective nature of the study; an audit can only evaluate the information that staff actually wrote in the medical record. Therefore, assessments that were carried out but not recorded will appear incomplete in the audit. Any clinical audit has the inherent limitation that lost values (data not available) can lead to measurement bias that may affect the results, regardless of the inclusion methodology and regular monitoring of the database. The evaluation of the possible interaction effect of more than two variables was not evaluated because it was difficult to interpret from a clinical point of view, given the limitations of being a retrospective study with unavailable data. Despite these limitations, we believe that this dataset represents the most comprehensive sample from respiratory clinics in Spain, providing real-world data on patients with high-risk COPD.

## Conclusions

This study provides information on the characteristics, interventions and medication prescription patterns in patients with high-risk COPD who were considered unstable because they reported moderate or severe exacerbations in the three months prior to the audited follow-up pulmonology visit. The results show that non-stability is present in one third

of patients with a high-risk level of COPD and that in more than half of the patients who are not stable at the visit, no changes are made in the pharmacological treatment.

## Supporting information

**S1 Table. Caracteristics asociated with oral therapies for COPD.**
(PDF)

**S2 Table. Factors associated with action taken during the visit in high risk level not stability patients.**
(PDF)

**S1 Fig. STROBE flow chart of the sampling process.**
(PDF)

**S1 Appendix. Participants in 2021 EPOCONSUL study.**
(PDF)

**S2 Appendix. The inclusion criteria and exclusion criteria.**
(PDF)

**S3 Appendix. Risk stratification according to GesEPOC.**
(PDF)

## Acknowledgments

The authors thank Chiesi for its support in carrying out the study.

## Author contributions

**Conceptualization:** Myriam Calle Rubio, Bernardino Alcázar-Navarrete, José Luis López-Campos, Marc Miravitlles, Juan José Soler-Cataluña, Juan Luis Rodriguez Hermosa.

**Data curation:** Myriam Calle Rubio, Manuel E. Fuentes Ferrer, Juan Luis Rodriguez Hermosa.

**Formal analysis:** Myriam Calle Rubio, Manuel E. Fuentes Ferrer, Juan Luis Rodriguez Hermosa.

**Funding acquisition:** Myriam Calle Rubio.

**Investigation:** Myriam Calle Rubio, Bernardino Alcázar-Navarrete, José Luis López-Campos, Marc Miravitlles, Juan José Soler-Cataluña, Juan Luis Rodriguez Hermosa.

**Methodology:** Myriam Calle Rubio, Bernardino Alcázar-Navarrete, José Luis López-Campos, Marc Miravitlles, Juan José Soler-Cataluña, Juan Luis Rodriguez Hermosa.

**Project administration:** Myriam Calle Rubio.

**Resources:** Myriam Calle Rubio.

**Supervision:** Myriam Calle Rubio.

**Validation:** Myriam Calle Rubio, Juan Luis Rodriguez Hermosa.

**Visualization:** Myriam Calle Rubio.

**Writing – original draft:** Myriam Calle Rubio, Juan Luis Rodriguez Hermosa.

**Writing – review & editing:** Myriam Calle Rubio, Bernardino Alcázar-Navarrete, José Luis López-Campos, Marc Miravitlles, Juan José Soler-Cataluña, Manuel E. Fuentes Ferrer, Juan Luis Rodriguez Hermosa.

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
