## [Decision Letter · Decision Letter 0]

PONE-D-25-12656Characteristics and actions in high-risk COPD in unstable patients: The EPOCONSUL auditPLOS ONE

Dear Dr. Rodriguez Hermosa,

Thank you for submitting your manuscript to PLOS ONE. After careful consideration, we feel that it has merit but does not fully meet PLOS ONE’s publication criteria as it currently stands. Therefore, we invite you to submit a revised version of the manuscript that addresses the points raised during the review process.

ACADEMIC EDITOR: The topis is of interest. However the manuscript needs a revision before it can be accepted. 

We look forward to receiving your revised manuscript.

Kind regards,

Thien Tan Tri Tai Truyen, M.D.

Academic Editor

PLOS ONE

Journal Requirements:

2. If any supporting files for review show as item type ‘other’ please change to item type ‘supporting info’ as the reviewer does not have access to these ’other’ files.

3. Thank you for stating the following financial disclosure: [This study has been promoted and sponsored by the Spanish Society of Pneumology and Thoracic Surgery (SEPAR).]. 

Reviewers' comments:

Reviewer's Responses to Questions

**Comments to the Author**

1. Is the manuscript technically sound, and do the data support the conclusions?

Reviewer #1: Partly

Reviewer #2: Yes

Reviewer #3: Yes

2. Has the statistical analysis been performed appropriately and rigorously? 

Reviewer #1: Yes

Reviewer #2: Yes

Reviewer #3: Yes

3. Have the authors made all data underlying the findings in their manuscript fully available?

Reviewer #1: Yes

Reviewer #2: Yes

Reviewer #3: Yes

4. Is the manuscript presented in an intelligible fashion and written in standard English?

Reviewer #1: Yes

Reviewer #2: Yes

Reviewer #3: Yes

5. Review Comments to the Author

Reviewer #1: This study provides valuable real-world insights into high-risk COPD management in Spain, highlighting key gaps in treatment and the importance of specialized care. However, causality cannot be established, and more work is needed to optimize therapy escalation and integrate biomarkers into decision-making.

Following are my comments:

1. Lack of standardized exacerbation criteria

COPD exacerbations are self-reported at follow-up visits, which introduces recall bias.

Future studies could use objective biomarkers (e.g., inflammatory markers, spirometry trends) to better define exacerbation severity.

2. Potential Confounding in Treatment Patterns

The study finds that oral therapies (macrolides, mucolytics, roflumilast) are used more often in unstable patients.

However, prescription patterns may reflect disease severity rather than therapeutic effectiveness.

A propensity-matched analysis could help determine whether treatment choices truly influence exacerbation risk.

4. Therapeutic Inertia: Lack of Detailed Exploration

More than 50% of unstable patients did not receive treatment changes, yet the reasons for this are not explored in depth.

Potential factors include: Physician hesitation due to comorbidities, Patient adherence issues, Lack of clear guidelines for therapy escalation.

Qualitative interviews with physicians could provide insights into clinical decision-making.

5. Need for Biomarker Integration

The study discusses peripheral eosinophilia as a potential marker for Th2 inflammation and treatment selection.

However, it does not analyze how eosinophil-guided therapy affected stability outcomes.

Future research should explore biomarker-driven treatment personalization.

6. Future Research Directions:

A. Longitudinal Studies

Follow-up studies should track exacerbation rates and lung function over time to assess the long-term impact of different treatment strategies.

B. Machine Learning for Risk Prediction

AI-based models could predict which high-risk patients are likely to deteriorate and personalize therapy accordingly.

C. Biomarker-Driven Treatment

Integrating blood eosinophils, sputum biomarkers, and genetics could help tailor COPD treatments more effectively.

Reviewer #2: It is a good study helping us understand the characteristics of high risk COPD patient and will helpful in taking the needed intervention for better management. It will helps in identifying the patients at risk for exacerbations and in turn will guide us in a better way to decrease the morbidity and mortility .

Reviewer #3: The study tackles persistent instability in high-risk patients, a clinical issue that is extremely pertinent to the therapy of COPD. One of its main advantages is the use of a national dataset in the real-world audit technique, which greatly expands on the body of current literature.

As the study is very essential to the investigation of COPD clearer figures ,Figures 1-4 visual segmentation and legends are unclear.

6. PLOS authors have the option to publish the peer review history of their article (what does this mean? ). If published, this will include your full peer review and any attached files.

**Do you want your identity to be public for this peer review?** For information about this choice, including consent withdrawal, please see our Privacy Policy .

Reviewer #1: No

Reviewer #2: **Yes: ** Jamal Akhtar

Reviewer #3: No

---

## [Author Response · Author response to Decision Letter 1]

15 May 2025

Journal: Plos One

Section: Original Research

Type of manuscript: Article

Revision required [PONE-D-25-12656] - [EMID:c3799199ce51f115]

Title: Characteristics and actions in high-risk COPD in unstable patients: The EPOCONSUL audit

Dear Editor Thien Tan Tri Tai Truyen

We would like to thank you. Below are the point-by-point answers to the remarks. You will find all the suggested changes tracked in a marked version of the original manuscript.

The full comments of the Editors are the following:

Journal Requirements:

Response:

We are grateful this comment.

Thank you very much. We have verified that the manuscript complies with the style requirements of PLOS ONE.

We have included the following funding statement in the cover letter ‘This study has been promoted and sponsored by the Spanish Society of Pneumology and Thoracic Surgery (SEPAR)]. The funders had no role in the study design, data collection and analysis, decision to publish or preparation of the manuscript’.

Reviewers' comments:

Reviewer's Responses to Questions

Reviewer Comments:

Reviewer #1:

This study provides valuable real-world insights into high-risk COPD management in Spain, highlighting key gaps in treatment and the importance of specialized care. However, causality cannot be established, and more work is needed to optimize therapy escalation and integrate biomarkers into decision-making.

Following are my comments:

1. Lack of standardized exacerbation criteria

COPD exacerbations are self-reported at follow-up visits, which introduces recall bias.

Future studies could use objective biomarkers (e.g., inflammatory markers, spirometry trends) to better define exacerbation severity.

RESPONSE: We welcome these comments, and include in the discussion this comment:

“The correct assessment of COPD exacerbations will be crucial to assess risk, the degree of clinical control in COPD and to plan therapeutic interventions. Currently the identification of exacerbating patients is based on clinical records and/or patient recall24. However, these are not always reported. Thus, several studies suggest that about half of exacerbations are not reported, although they are associated with a worsening quality of life and an increased risk of subsequent hospitalisation2526. On the other hand, it should be remembered that definitions of COPD exacerbations are mainly based on clinical presentation, which can be non-specific and lead to misdiagnosis. The use of biomarkers could provide a more objective and specific definition of COPD exacerbations and allow for a more accurate assessment of progression or response to treatment. Future research is needed to establish a better definition of COPD exacerbations through the use of biomarkers to improve diagnostic accuracy, more personalised treatment and potentially improve clinical outcomes and reduce healthcare costs.”

2. Potential Confounding in Treatment Patterns

The study finds that oral therapies (macrolides, mucolytics, roflumilast) are used more often in unstable patients.

However, prescription patterns may reflect disease severity rather than therapeutic effectiveness.

A propensity-matched analysis could help determine whether treatment choices truly influence exacerbation risk.

RESPONSE: We welcome these comments, which we include in the discussion :

“In our analysis, being a cross-sectional observational study, prescription patterns may reflect disease severity rather than therapeutic effectiveness. Furthermore, it should be borne in mind that the clinical presentation of COPD is highly variable, and that other factors associated with lack of clinical control and stability include associated comorbidities, adherence, among others.”

3. Therapeutic Inertia: Lack of Detailed Exploration

More than 50% of unstable patients did not receive treatment changes, yet the reasons for this are not explored in depth.

Potential factors include: Physician hesitation due to comorbidities, Patient adherence issues, Lack of clear guidelines for therapy escalation.

Qualitative interviews with physicians could provide insights into clinical decision-making.

RESPONSE We appreciate these considerations in relation to non-performance at the visit. Our study is a retrospective audit of clinical records, where conditioning factors such as knowledge, experience of the professional attending the consultation, availability of time or means, or clinical considerations of the patient could not be evaluated.

We include a commentary in the discussion as a line of interest for further study that may help to identify limitations in the implementation of good clinical practice guidelines and conditioning factors in clinical practice.

“Little is known about the reasons for the lack of action when treatment goals are not met. Studies have shown that physicians are willing to take actions when they deem them appropriate, but there is a disconnection between their perception and the actual degree of control of COPD in uncontrolled patients, suggesting that physicians do not feel the need to switch up therapies or increase doses if the patient is showing positive signs of improvement or maintenance34. Studies are needed to analyze the reasons for the non-implementation of clinical practice guidelines and the conditioning factors in clinical practice.”

4. Need for Biomarker Integration

The study discusses peripheral eosinophilia as a potential marker for Th2 inflammation and treatment selection.

However, it does not analyze how eosinophil-guided therapy affected stability outcomes.

Future research should explore biomarker-driven treatment personalization.

RESPONSE: We appreciate these considerations. We include a commentary in the discussion as a line of interest for further study

““The use of specific biomarkers in the future could allow for a better diagnostic approach in characterising exacerbations but could also offer a more personalised treatment and potentially improve clinical outcomes.”

6. Future Research Directions:

A. Longitudinal Studies

Follow-up studies should track exacerbation rates and lung function over time to assess the long-term impact of different treatment strategies.

B. Machine Learning for Risk Prediction

AI-based models could predict which high-risk patients are likely to deteriorate and personalize therapy accordingly.

C. Biomarker-Driven Treatment

Integrating blood eosinophils, sputum biomarkers, and genetics could help tailor COPD treatments more effectively.

RESPONSE: We appreciate these considerations. We include future directions

“Artificial intelligence-based models can significantly enhance the prediction of deterioration in high-risk COPD patients and personalize therapy. AI algorithms, including machine learning and deep learning, can analyze diverse data types such as clinical records, patient-reported outcomes, environmental factors, and genetic information to predict exacerbations and disease progression. Models using support vector machines, boosting, and deep neural networks have shown promise in predicting COPD exacerbations with high accuracy 39”

Reviewer #2: It is a good study helping us understand the characteristics of high risk COPD patient and will helpful in taking the needed intervention for better management. It will helps in identifying the patients at risk for exacerbations and in turn will guide us in a better way to decrease the morbidity and mortility .

RESPONSE: We sincerely thank you for your positive feedback and encouraging comments. We greatly appreciate your review and support of our work.

Reviewer #3: The study tackles persistent instability in high-risk patients, a clinical issue that is extremely pertinent to the therapy of COPD. One of its main advantages is the use of a national dataset in the real-world audit technique, which greatly expands on the body of current literature.

As the study is very essential to the investigation of COPD clearer figures ,Figures 1-4 visual segmentation and legends are unclear.

RESPONSE: We sincerely thank you for your positive feedback and encouraging comments. We greatly appreciate your review and support of our work.

We have clarified the data in the figures 1A, 1B and 1C

---

## [Decision Letter · Decision Letter 1]

PONE-D-25-12656R1Characteristics and actions in high-risk COPD in unstable patients: The EPOCONSUL auditPLOS ONE

Dear Dr. Rodriguez Hermosa,

Thank you for submitting your manuscript to PLOS ONE. After careful consideration, we feel that it has merit but does not fully meet PLOS ONE’s publication criteria as it currently stands. Therefore, we invite you to submit a revised version of the manuscript that addresses the points raised during the review process.

**ACADEMIC EDITOR: A minor revision is needed to address final comments from reviewers. ** ==============================

We look forward to receiving your revised manuscript.

Kind regards,

Thien Tan Tri Tai Truyen, M.D.

Academic Editor

PLOS ONE

Journal Requirements:

Reviewers' comments:

Reviewer's Responses to Questions

**Comments to the Author**

1. If the authors have adequately addressed your comments raised in a previous round of review and you feel that this manuscript is now acceptable for publication, you may indicate that here to bypass the “Comments to the Author” section, enter your conflict of interest statement in the “Confidential to Editor” section, and submit your "Accept" recommendation.

Reviewer #1: All comments have been addressed

Reviewer #3: All comments have been addressed

2. Is the manuscript technically sound, and do the data support the conclusions?

Reviewer #1: (No Response)

Reviewer #3: Yes

3. Has the statistical analysis been performed appropriately and rigorously? 

Reviewer #1: (No Response)

Reviewer #3: Yes

4. Have the authors made all data underlying the findings in their manuscript fully available?

Reviewer #1: (No Response)

Reviewer #3: Yes

5. Is the manuscript presented in an intelligible fashion and written in standard English?

Reviewer #1: (No Response)

Reviewer #3: Yes

6. Review Comments to the Author

Reviewer #1: (No Response)

Reviewer #3: Recall bias is introduced by the operational definition of "non-stability," which is predicated on patient-reported exacerbations within three months.

The finding that 56% of unstable patients had no change in therapy warrants deeper analysis.

Although important clinical variables are adjusted for in multivariate models, it is uncertain whether interaction effects.

The discussion highlights the importance of Th2 inflammation and eosinophils, but it does not completely incorporate their significance into the findings or clinical implications.

7. PLOS authors have the option to publish the peer review history of their article (what does this mean? ). If published, this will include your full peer review and any attached files.

**Do you want your identity to be public for this peer review?** For information about this choice, including consent withdrawal, please see our Privacy Policy .

Reviewer #1: No

Reviewer #3: No

---

## [Author Response · Author response to Decision Letter 2]

10 Jun 2025

Journal: Plos One

Section: Original Research

Type of manuscript: Article

Revision required [PONE-D-25-12656] - [EMID:c3799199ce51f115]

Title: Characteristics and actions in high-risk COPD in unstable patients: The EPOCONSUL audit

Dear Editor Thien Tan Tri Tai Truyen

We would like to thank you. Below are the point-by-point answers to the remarks. You will find all the suggested changes tracked in a marked version of the original manuscript.

Reviewers' comments:

Reviewer's Responses to Questions

Reviewer #3:

6. Review Comments to the Author

Reviewer #3: Recall bias is introduced by the operational definition of "non-stability," which is predicated on patient-reported exacerbations within three months.

RESPONSE: Taking into account your recommendation, we have clarified the definition of “no stability”, including this additional information

In abstract:” Objective: to evaluate the clinical characteristics of high-risk COPD patients considered unstable for having had moderate or severe exacerbations of COPD in the three months prior to the audited review visit based on information extracted from the medical record documenting health interactions prior to the visit.”

In the methodology section:

“Non-stability (non-stable patient) was defined on the basis of having moderate or severe exacerbations of COPD in the three months prior to the audited review visit. All data were verified by clinical investigators with access to the original documentation. Data were systematically extracted from medical records documenting health interactions prior to the audited visit to ensure completeness and consistency with records made at the audited visit.”

Reviewer #3:

The finding that 56% of unstable patients had no change in therapy warrants deeper analysis.

RESPONSE: Taking into account your recommendation, we have include in results table 5: Factors associated with no change in therapy during the visit in high risk level not stability patients

We have and include in the discussion this comment: “In our analysis, few clinical characteristics were associated with the dependent variable no change in COPD treatment at the visit in a patient with high-risk COPD who reported exacerbations at the visit. In the multivariate logistic regression model, only being treated in a university hospital (OR 3.08, 95% CI 1.71-5.53; p<0.001) was found as an associated factor, being less likely not to make changes in the active smoker (OR 0.63, 95% CI 0.44-0.90; p=0.011), and the presence of chronic bronchial infection (OR 0.50, 95% CI 0.33-0.76; p=0.001).”

Reviewer #3: Although important clinical variables are adjusted for in multivariate models, it is uncertain whether interaction effects.

RESPONSE: Taking into account your recommendation. This aspect is included as a limitation.

“the evaluation of the possible interaction effect of more than two variables was not evaluated because it was difficult to interpret from a clinical point of view, given the limitations of being a retrospective study with unavailable data..”

we have not evaluated the interaction from a statistical point of view because, being a retrospective study whose source is the medical records, the evaluation of the possible interaction effect of more than two variables would be difficult to interpret from a clinical point of view.

Reviewer #3: The discussion highlights the importance of Th2 inflammation and eosinophils, but it does not completely incorporate their significance into the findings or clinical implications.

RESPONSE: Taking into account your recommendation, we have and include in the discussion this comment:

“In our population of unstable high-risk patients, 85% had a frequent exacerbator phenotype and 67% were on maintenance treatment with triple therapy. Of these, 23% had peripheral blood eosinophil counts ≥300 cells/μL, being a priori a population in which treatment optimization should be considered, assessing escalation to triple therapy and/or could benefit from biologic therapy. Evidence suggests that 20-40% of COPD patients have Th2 airway inflammation as measured by sputum eosinophilia �37,38]. This endotype is associated with increased risk of exacerbations. Elevated blood eosinophil counts (BEC), typically ≥300 cells/μL, serve as a practical biomarker, identify patients more likely to benefit from emerging biologics. Recent trials demonstrate that targeting T2 pathways produces significant reductions in exacerbations and improves lung function in COPD patients who persist with exacerbations despite optimization of inhaled therapy �38], reflecting a move toward precision medicine in this heterogeneous disease. In the near future, artificial intelligence-based models may help to identify this group of patients earlier in order to provide more personalized medicine for high-risk COPD patients. AI algorithms, including machine learning and deep learning, can analyze various types of data, such as medical records, patient-reported outcomes, environmental factors and genetic information, to predict exacerbations and disease progression.”

---

## [Editor Report · Decision Letter 2]

Characteristics and actions in high-risk COPD in unstable patients: The EPOCONSUL audit

PONE-D-25-12656R2

Dear Dr. Rodriguez Hermosa,

We’re pleased to inform you that your manuscript has been judged scientifically suitable for publication and will be formally accepted for publication once it meets all outstanding technical requirements.

Kind regards,

Thien Tan Tri Tai Truyen, M.D.

Academic Editor

PLOS ONE
---

## [Editor Report · Acceptance letter]

PONE-D-25-12656R2

PLOS ONE

Dear Dr. Rodriguez Hermosa,

I'm pleased to inform you that your manuscript has been deemed suitable for publication in PLOS ONE. Congratulations! Your manuscript is now being handed over to our production team.

Kind regards,

on behalf of

Dr. Thien Tan Tri Tai Truyen

Academic Editor

PLOS ONE